# A New NILM System Based on the SFRA Technique and Machine Learning

**DOI:** 10.3390/s23115226

**Published:** 2023-05-31

**Authors:** Simone Mari, Giovanni Bucci, Fabrizio Ciancetta, Edoardo Fiorucci, Andrea Fioravanti

**Affiliations:** Dipartimento di Ingegneria Industriale e dell’Informazione e di Economia, Università dell’Aquila, 67100 L’Aquila, Italy; giovanni.bucci@univaq.it (G.B.); fabrizio.ciancetta@univaq.it (F.C.); edoardo.fiorucci@univaq.it (E.F.); andrea.fioravanti@univaq.it (A.F.)

**Keywords:** machine learning (ML), nonintrusive load monitoring (NILM), smart home, support vector machine (SVM), sweep frequency response analysis (SFRA)

## Abstract

In traditional nonintrusive load monitoring (NILM) systems, the measurement device is installed upstream of an electrical system to acquire the total aggregate absorbed power and derive the powers absorbed by the individual electrical loads. Knowing the energy consumption related to each load makes the user aware and capable of identifying malfunctioning or less-efficient loads in order to reduce consumption through appropriate corrective actions. To meet the feedback needs of modern home, energy, and assisted environment management systems, the nonintrusive monitoring of the power status (ON or OFF) of a load is often required, regardless of the information associated with its consumption. This parameter is not easy to obtain from common NILM systems. This article proposes an inexpensive and easy-to-install monitoring system capable of providing information on the status of the various loads powered by an electrical system. The proposed technique involves the processing of the traces obtained by a measurement system based on Sweep Frequency Response Analysis (SFRA) through a Support Vector Machine (SVM) algorithm. The overall accuracy of the system in its final configuration is between 94% and 99%, depending on the amount of data used for training. Numerous tests have been conducted on many loads with different characteristics. The positive results obtained are illustrated and commented on.

## 1. Introduction

The goal of energy saving within modern smart homes and energy management systems is pursued by monitoring and controlling household parameters, such as lighting and home temperature [1]. This need has led to a significant increase in attention to nonintrusive load-monitoring (NILM) systems.

Among the energy-monitoring systems, those based on the NILM technique represent one of the most relevant solutions. The total energy consumption of users is monitored and the consumption of each individual load is identified. For this purpose, the measurements of current and voltage are carried out, or often of the current alone; the data collected are then processed with a so-called “disaggregation” algorithm. The main advantages of the nonintrusiveness are the simplicity and cost-effectiveness of installation. Therefore, systems of this type are useful for both consumers and utility companies when analyzing the use and costs of electricity.

In the early 1990s, the first NILM system was proposed [2]. Since then, more advanced algorithms have enabled a significant improvement in energy-unbundling systems.

This is especially true over the past decade, which has seen a significant increase in interest in this topic.

The first NILM systems detected events and classified the various loads using traditional algorithms [2,3]. The most modern, however, use artificial intelligence algorithms, in particular, through machine-learning (ML) techniques. For example, in some studies [4,5], the energy disaggregation problem has been reformulated as an adaptive filtering problem; refs. [6,7] propose model-driven NILM systems and the works proposed in other studies [8,9,10] are based on hidden Markov chains, while others [11,12,13,14,15] use artificial neural networks. The latter types of algorithms learn from the data provided and can perform certain tasks. Therefore, ML algorithms continue to improve over time by learning from data with minimal human intervention [16].

With regard to the NILM problem, these systems process the active power—and sometimes also the reactive power—absorbed by the monitored system [16]. However, NILM systems have been developed based on transient rather than stationary characteristics or the analysis of other quantities, which differ due to belonging to different domains (time or frequency).

In this sense, the sampling frequency is a fundamental parameter used to define the extractable information. Low-frequency time series were processed to evaluate steady-state characteristics. On the other hand, time series at other frequencies were processed to obtain information about the startup and shutdown transients to be able to discriminate the loads through their dynamic parameters (overshoot, rise time, etc.) or by characterizing appliances based on the pulses produced on the power line. Other attempts have been made by processing the trajectories drawn on the V-I plane.

Today, the division of NILM systems into event-driven and non-event-driven systems is the most widely used division and is the best for defining the state of the art of these systems.

The former involves the detection of an event (understood as an appliance turning on, turning off, or switching to a different consumption state) and then classifying it based on the features associated with the appliance that caused it. This type of approach can therefore be divided into three basic steps: event detection, feature extraction, and load identification. In particular, the last step is performed in most cases using ML algorithms that work well as classification systems. Numerous supervised ML algorithms have been proposed in the literature, including K Nearest Neighbor (KNN) [17], naïve Bayes [18], Decision Tree (DT) [19], Support Vector Machine (SVM) [20], Principal Component Analysis (PCA) [21], and Artificial Neural Network (ANN) [22,23]. Finally, unsupervised [24] and semi-supervised learning algorithms [25], as well as those related to graph signal processing [26], have also been proposed.

On the other hand, non-event-based systems are NILM systems that do not have an event-detection phase. In these cases, the concept of the “signature”/“features” of an appliance is also lost, as the only feature used by the models is the aggregate power profile. They use a window of samples of the aggregate signal (therefore time series data) as input; the samples are processed continuously without waiting for the occurrence of events. For this reason, this type of system is particularly suitable for low-frequency signals. Indeed, it was developed precisely to allow the processing of signals acquired with reduced frequencies, for which the detection of events is more difficult. In some cases, the disaggregation problem is formulated as a blind source separation (BSS) problem—that is, the problem of recovering a signal from a set of mixed signals. Numerous approaches have been proposed in the last decade, the most significant being those based on Combinatorial Optimization [27], Discriminative Sparse Coding [28], Hidden Markov Model Approaches [9,29,30,31,32,33,34,35], and Deep Learning (DL) [36,37,38,39,40,41,42,43].

NILM systems are used in a wide range of applications. Among these, very promising are the applications in Ambient Assisted Living, i.e., systems that make it possible to meet the needs of elderly or disabled users, allowing them to live independently [44]. In fact, knowing the changes in the status of the various appliances in a relatively short time, it is possible to infer the Activities of Daily Living (ADL) of the occupants.

This paper presents a measurement system for nonintrusive monitoring (it does not require modifications to the electrical system) based on the injection of a variable frequency sinusoidal signal and the characterization of the system based on the response to it. This technique is called Sweep Frequency Response Analysis (SFRA) and is widely used in diagnostics and fault-finding in transformers and electric motors.

The proposed solution is very different from the other solutions proposed in the literature, which provide for the analysis of time-varying electrical signals through different approaches. Following these approaches, the focus is generally on transients of the absorbed current, which indicate a change in the connection state. The measurement of the current in static conditions does not allow the identification of active devices, except in very special simple cases.

The approach proposed in this paper makes it possible to identify which appliances are inserted through a measurement performed in static conditions (not in the connection/disconnection transient). It allows for the detection of a sort of signature that is unique and independent of the absorbed current. This approach, as illustrated below, allows us to overcome the typical problems of NILM systems in identifying multi-state or continuous variable load household appliances (or, in general, electrical loads).

All the SFRA apparatuses available on the market can only work on single devices that are switched off and disconnected from the grid. The SFRA system proposed in this article can operate online [45], thus allowing it to extend its operating range to systems for continuous diagnostics on devices while supplied by the mains; no functioning interruptions or disconnection operations are needed for the standard SFRA apparatuses.

The proposed system is based on a machine-learning algorithm, the Support Vector Machine (SVM), which is capable of determining the status of individual household appliances starting from the measurement obtained by the SFRA system. It was installed on a home test system and acquired and processed the data locally. 

Extensive measurements were made in order to verify the operational characteristics. The results obtained from field applications are also included and discussed.

## 2. Frequency Response Analysis of Household Appliances

SFRA has been successfully used to perform diagnostics on the windings of electric machines during the production process [46,47]. An electric machine can be considered a complex electrical network of capacitances, inductances, and resistors. As shown in Figure 1, the SFRA instrument injects a sinusoidal excitation voltage (the typical amplitude is 10 Vpp) with a continuously increasing frequency into one end of the transformer winding and measures the signal returning from the other end. This test is conducted with the machine disconnected from the power line. More details are reported in another study [46].

The comparison of input and output signals generates a frequency response, which can be compared with reference data. Degradation of the insulating materials or a change in the shape of the windings will result in a change in the RLC components of the network and, consequently, in the frequency response curve. Faults can therefore be detected by processing correlation indices between different curves.

In the proposed application, shown in Figure 2, the SFRA technique is applied to the electrical system supplied by the mains in order to obtain a signature that allows for discriminating different power supply conditions of a domestic system. The applied signal and the output signal, between the terminal of the neutral conductor and the ground, are acquired and processed by the system. The proposed measurement system can therefore be conveniently installed on a standard domestic socket.

A low-voltage (±5 Vpp) sinusoidal signal with variable frequency (from 2 kHz to 1.5 MHz) is superimposed on the supply voltage (240 Vrms and 50 Hz) and applied between the power phase conductor terminal and ground.

The signal generator is coupled to the network by means of a band-pass filter that allows only the passage of the test signal. The two input channels of the measurement circuit are also decoupled from the power supply by two other band-pass filters. The filters block both the fundamental frequency (50 Hz) and the harmonic components (up to 2 kHz) [48].

As the first part of the work, the system’s response was evaluated over a fairly wide frequency range and by acquiring a sufficiently high number of points.

The frequency response was obtained by injecting a signal generated at 100 MS/s. In order to optimize the memory, the sampling frequencies to acquire both applied and output signals were adapted according to the frequency to be analyzed. In detail, the sampling frequency was chosen as being equal to 25 times the analyzed frequency. To obtain a better resolution, the FFT was performed by fixing a frequency bin at the frequency of the generated sinusoid. The FFT was also performed on the output signal and the sample at the same bin was considered.

A Hanning window with a width equal to the acquisition time (corresponding to 64 cycles of the generated frequency) was used to process the FFT. Downstream of the FFT processing, the system calculated the $V_{out}/V_{in}$ ratio. For example, the 1 kHz response is achieved by injecting a 1 kHz sinusoidal signal generated at a frequency of 100 MS/s. The applied signal and the output signal were sampled at a sampling rate of 25 × 1000 = 25,000 Hz. A time window of (1/1000) × 64 = 0.064 s was considered for the processing of the FFTs, corresponding to 1600 samples. This process was repeated for all the frequencies of interest. The block diagram of the LabVIEW code is shown in Appendix A.

In order to evaluate the validity of the signature for different frequency ranges, four sub-bands were defined:(1)2–10 kHz;(2)10–100 kHz;(3)100 kHz–1 MHz;(4)1–1.5 MHz.

For each sub-band, 200 points were initially acquired. These sub-divisions were obtained considering the possible response to this type of excitation signal. Figure 3 schematically shows the installation of the SFRA system in the test system. From the knowledge in the literature about SFRA [48], the low-frequency response (2–10 kHz) is characterized by an ohmic-inductive behavior in which the characteristics of the grid upstream of the system are predominant; therefore, the contribution of the loads is usually not significant. The medium-frequency response (10 kHz–1 MHz) is characterized by resonance phenomena. As this band is generally the most interesting in terms of the effect of loads on the response, it has been split into two sub-bands to increase resolution. The high-frequency response (1–1.5 MHz) is characterized by capacitive effects due both to the network and the user loads and the connection of the measuring instrument itself, which generally determine a poor reproducibility of the measurement.

The sinusoidal test signal introduces no problems to the system. This is essentially due to the reduced amplitude of the test signal with respect to the line voltage (1.54%), which is fully within the limits imposed by the standard [49].

During the tests, it was verified that the signal does not create problems in intelligent automation systems operating with conveyed waves [50]. This is also because these systems adopt sophisticated signal-modulation algorithms that encode the data transmitted with different sub-carriers or that widen the transmission band (Spread Spectrum), obtaining a better resistance to interference and noise. Other systems adopt Orthogonal Frequency Division Multiplexing (OFDM) modulation techniques, which are even more effective.

Several tests were performed at a residential test facility. A wide variety of loads were taken into consideration, powering them individually or simultaneously and under different working conditions:(1)Hairdryer;(2)Microwave oven;(3)Lamp;(4)Laptop;(5)Induction hob;(6)Heater;(7)Drill;(8)TV.

Figure 4 shows the frequency response of these appliances when powered individually. The measurements were conducted in 24 different power supply scenarios, as summarized in Table 1. It is important to note that Scenario 1 represents the case in which none of the appliances was powered (condition indicated with “Open Circuit” in Figure 1). Scenarios 2 to 9 represent the single power supply conditions of household appliances. Scenarios 10 to 24 represent the simultaneous power conditions.

To support an objective evaluation, Figure 5 shows the lower and upper envelopes of the traces obtained in the presence and absence of each of the eight considered appliances, obtained following the measurements performed for the different scenarios. Measurements were performed for each of the 24 scenarios reported in Table 1, thus obtaining 24 SFRA traces. For each envelope (related to each appliance), the traces were divided into two groups according to the presence or absence of the appliance in the power supply scenario. The envelopes were then obtained by considering the maximum and minimum values of each of the two groups for each frequency bin. From these envelopes, it is immediately evident that the contribution of the low-frequency measurement (2–10 kHz) is not influenced by the different load configurations; therefore, in the rest of the work, we will only refer to the other three sub-bands.

These traces were used as inputs to a machine-learning-based classification algorithm, the Support Vector Machine (SVM), to determine the correct combination of powered appliances. A NILM system based on this type of input is easy to install, as it can be connected to a standard domestic socket, such as any household appliance. Traditional NILM systems, on the other hand, measure the aggregate power upstream of the plant and therefore require a more difficult installation. 

The measurement obtained represents the transfer function of the equivalent RLC circuit [23]. Therefore, the result is mainly influenced by the physical characteristics of the appliances rather than by their power absorption. This represents a great advantage for the discrimination of multi-state or continuously variable load appliances (such as drills) whose identification is often critical for systems based on the analysis of power consumption. 

The transfer function is minimally influenced by the choice of the socket in which to install the measuring system. Tests were carried out in all the sockets shown in Figure 3; all of the possible positions of the instrument on the various sockets allow the maximum reproducibility of the measurement. Regardless, the instrument is meant to be used on a single socket. The proposed algorithm is described in Section 3.

## 3. Machine-Learning Systems

Machine learning is the field of study that allows computers to learn without being explicitly programmed [51]. Unlike traditional programming, which provides a list of more or less complex rules defined by the programmer to obtain certain outputs, machine learning automatically learns patterns and correlations to solve extremely complex problems. In problems where existing solutions require a lot of manual adjustments or long lists of rules, a machine-learning algorithm can often simplify the code and achieve better performance. Sometimes they allow us to find solutions to problems that otherwise would not be solved through traditional approaches. These algorithms are used to process large amounts of data in order to discover patterns that are not immediately apparent. They are also used in situations where the algorithm needs to dynamically adapt to new patterns in the data or when the data itself is generated as a function of time, such as stock price prediction; in this case, we speak of online learning.

Machine-learning algorithms can be classified into supervised learning, unsupervised learning, semi-supervised learning, and reinforcement learning. This classification is made in relation to the quantity of data available during the training phase and the type of supervision during the training.

Specifically, in supervised learning, the training data provided to the algorithm include desired solutions called labels. Supervised learning solves two types of problems: classification and regression.

Classification is the problem of cataloging data into two or more classes; so, by providing input to the machine-learning system, it must return its class of belonging. 

On the other hand, regression interpolates data to associate two or more features with each other. By providing the algorithm with an input feature, the regressor returns the other feature. A system of estimating the price of houses starting from features, such as size, number of rooms, and area, is a regression system. 

The most popular supervised-learning algorithms are k-Nearest Neighbors, linear regression, logistic regression, Support Vector Machine (SVM), Decision Trees, Random Forests, and Neural Networks.

The NILM problem can be set up either as a regression problem—for example, when the algorithm is called to estimate the power absorbed by the single appliance starting from the aggregate power measurement [52]—or as a classification problem [53], as in the case in which starting from the aggregate power measurement is necessary to determine which appliances are powered and which are not.

The system proposed in this manuscript solves a multi-label classification problem since, starting from an SFRA trace, it is possible to identify several powered appliances simultaneously. The algorithm used is the SVM; the system configuration and its operation are illustrated in the following paragraphs.

### 3.1. Support Vector Machine

A SVM is one of the most popular models in machine learning, as it is very powerful and versatile [51]. SVMs are best suited for classifying complex but small- to medium-sized datasets. While classic classification algorithms discriminate based on characteristics common to each class, the SVM algorithms build the model based on the most difficult samples to discriminate, i.e., the most similar samples belonging to different classes. In this sense, the only samples used in the construction of the model are called support vectors. The other samples are therefore useless.

Based on the support vectors, the algorithm finds the optimal hyperplane that separates them, which can then be used to discriminate new samples. In other words, adding more formation samples far from the hyperplane (therefore not particularly complex to classify) will not affect the decision boundary, which will be completely determined by the samples located at the edge of the hyperplane.

Consider a case in which the samples to be classified are defined by only two features.

This case can be represented on a two-dimensional plane, as shown in Figure 6. A SVM algorithm looks for the line capable of maximizing the margin between the most similar samples belonging to different classes, i.e., the support vectors.

Consider a linear classification problem in which n-dimensional inputs X are divided into two classes $y\in \left\{-1,1\right\}$. The classifier can be formulated as follows:(1)$$f_{\left(x\right)}=w^{T}\phi_{\left(x\right)}+b,$$
where w is the vector of weights, b is the bias, and $\phi_{\left(x\right)}$ is the feature space of the inputs. The sign of f(x) will be the output $y_{i}$ of the classification.

Since the inputs are linearly separable, it will be possible to choose several linear decision boundaries, each of which will not produce classification errors in the training data.

Training a SVM model positions the boundary to maximize the margin—that is, the distance from the hyperplane to the nearest data point in either class. More specifically, we want to optimize the following objective function:(2)$$\underset{w,b}{\max}\underset{i}{\min}dist(x_{i},w,b)\;|\;\forall i\;y_{i}(w^{T}\phi_{\left(x_{i}\right)}+b)\geq 0,$$
where dist(x,w,b) is the Euclidean distance from the feature point $\phi_{\left(x\right)}$ to the hyperplane defined by w and b. With this objective function, the distance from the decision boundary $w^{T}\phi_{\left(x\right)}+b=0$ to the nearest point i is maximized. The constraints force finding a decision boundary that correctly classifies all the training data. In other words, for the classifier, a correct training point $y_{i}$ and $w^{T}\phi_{\left(x_{i}\right)}+b$ must have the same sign, in which case their product must be positive.

It is known from Euclidean geometry that the distance between the point $\phi_{\left(x_{i}\right)}$ and the hyperplane $w^{T}\phi_{\left(x\right)}+b=0$ can be defined as $\frac{{|w}^{T}\phi_{\left(x_{i}\right)}+b|}{\left||w|\right|}$. Since $y_{i}$ is the sign of $f_{\left(x_{i}\right)}$, it can be written as follows:(3)$$\underset{w,b}{\max}\underset{i}{\min}\frac{{y_{i}(w}^{T}\phi_{\left(x_{i}\right)}+b)}{\left||w|\right|}\;|\;\forall i\;y_{i}(w^{T}\phi_{\left(x_{i}\right)}+b)\geq 0,$$

We can observe that, due to the normalization of ||w|| in (3), the scale of w is arbitrary in this objective function. That is, if w and b are multiplied by a real scalar α, the factors of α in the numerator and denominator will cancel each other out. Now, suppose we choose the scale so that the point closest to the hyperplane, *x_i_* satisfies ${y_{i}(w}^{T}\phi_{\left(x_{i}\right)}+b)=1$. With this assumption, the $\underset{i}{min}$ in Equation (3) becomes redundant and can be removed. The objective function and constraint can be rewritten as:(4)$$\underset{w,b}{\max}\frac{1}{\left||w|\right|}\;|\;\forall i\;y_{i}(w^{T}\phi_{\left(x_{i}\right)}+b)\geq 0,$$

Finally, we convert the problem into a quadratic program (QP). In this way, the objective function is quadratic in the unknowns and all constraints are linear in the unknowns. A QP has a single global minimum, which can be found efficiently with current optimization packages [54].
(5)$$\underset{w,b}{\max}\frac{1}{2}{\left||w|\right|}^{2}\;|\;\forall i\;y_{i}(w^{T}\phi_{\left(x_{i}\right)}+b)\geq 0,$$

However, not all classification problems are linear; in fact, in some cases, it is not possible to separate the classes with a straight line; therefore, we speak of non-linear classification. The kernel trick [55] solves non-linear classification problems with SVM algorithms.

In more detail, a polynomial kernel was used to determine the presence, or absence, of an appliance starting from the SFRA traces. Using a polynomial kernel means determining similarity, not only by processing the features of the input samples but also by their combinations, as shown in Figure 7.

Moreover, in real scenarios, data belonging to different classes overlap. As a result, it will not be possible to satisfy all the constraints in (5). One way to deal with this problem and still train useful classifiers is to relax some constraints by introducing so-called slack variables [56]. Normally, a Lagrangian transformation addresses the optimization problem, which allows the constrained optimization problem expressed in (5) to be reformulated into a non-constrained optimization problem.

The Lagrangian for the SVM objective function in (5), with Lagrange multipliers $a_{i}\geq 0$, is:(6)$$L_{\left(a_{1:N}\right)}=\sum a_{i}-\frac{1}{2}\sum_{i}^{\;}\sum_{j}^{\;}a_{i}a_{j}y_{i}y_{j}k_{(x_{i},x_{j}),}$$
where $k_{\left(x_{i},x_{j}\right)}$ is called a kernel function. For example, if we used the basic linear features, i.e., $\phi_{\left(x\right)}=x$, then $k_{\left(x_{i},x_{j}\right)}={x_{i}}^{T}x_{j}$. Instead, because a polynomial kernel has been chosen in the implemented SVM classifier, it will be defined as:(7)$$k_{\left(x_{i},x_{j}\right)}={\left(a+{x_{i}}^{T}x_{j}\right)}^{b},$$

### 3.2. The Proposed Structure

In the proposed system, the input is the trace obtained from the SFRA system; thus, each point of the trace represents a feature of the SVM. The algorithm must have a number of input functions equal to the number of bins of the measured frequency response.

The problem is also attributable to a multi-label classification problem, where a single sample can belong to multiple defined classes, unlike in multi-class classification, where each sample can uniquely belong to only one class.

In fact, the purpose of the system is to determine the status (ON or OFF) of the appliances. This means that the number of classes is equal to that of the appliances and the belonging of an SFRA trace to a certain class will indicate the ON state of that appliance. A single SFRA trace must therefore be able to be associated with multiple classes (or labels), as the system must be able to recognize the loads even under simultaneous power supply conditions. SVMs are not natively capable of performing multi-class or multi-label classifications since, as explained above, a SVM defines a hyperplane that separates classes equidistantly in order to guarantee the maximum margin. When the number of classes rises to three or more, thus passing from a binary classification to multi-class, it is possible to guarantee equidistance only between two of the classes, discarding this property with all the other classes.

To solve this classification problem, which involves assigning multiple labels to an instance, we converted it to multiple binary classification problems. A SVM was therefore associated with each household appliance, performing a binary classification in order to determine its ON or OFF status, starting from the SFRA trace. The proposed structure is shown in Figure 8.

## 4. Experimental Results

As part of the development phase, the proposed algorithm was implemented and tested to evaluate its performance with real data.

### 4.1. The Proposed System Setup

As explained in Section 2, the SFRA technique was performed by plugging the instrument into a standard household socket. As previously discussed, the input signal is a variable frequency sinusoidal signal applied between the phase conductor terminal and ground, while the output signal is the measured signal between the neutral conductor terminal and ground. Both signals are acquired and processed. Figure 9 shows the measurement system used.

The measurement system must be connected to the test system by means of cables with suitable bandwidth and the same characteristic impedance of the generator to avoid reflection and signal mismatch and to improve the sensitivity, repeatability, and reliability of the measurement. 

The input signal and related acquisition for the SFRA were performed using the Digilent Analog Discovery 2 NI Edition card with a BNC adapter.

The control system was developed using LabVIEW and run on a PC; this software automatically programs the Discovery FPGA at startup, with a configuration file designed to implement the measurement application. Once programmed, the integrated FPGA communicates with the PC via a USB 2.0 connection. The PC enables the creation of the user interface to access the data and process them in the experimental phase. A final NILM system can bypass the PC by integrating post-processing directly into the system.

The Discovery FPGA has a ±25 V input range, a 14-bit resolution, a 100 MS/s sampling frequency, and a 30 MHz bandwidth. It is equipped with an arbitrary function generator with an output range of ±5 V, a bandwidth of 20 MHz, and a sampling rate of 100 MS/s.

For appropriate interfacing with the network, the instrument is equipped with a coupling circuit for each of the three channels (one for generation and two for acquisition), as shown in Figure 9. The coupling circuit includes a third-order Butterworth filter with a flat passband and high attenuation outside the desired frequency range. The generation section and acquisition section coupling circuits both involve a 50 Ω resistor in series and parallel, respectively, to allow impedance adaptation. In addition, all coupling circuits are provided with a high-voltage ac blocking capacitor, connected in series with a 1:1 pulse transformer. The features of the filters developed for the SFRA apparatus are shown in Figure 10 and Figure 11.

In order to avoid unwanted over-voltages due to resonance phenomena at high frequencies, the amplitude of the applied signal must not exceed a few volts (5 Vpp in the present case). The accuracy of the adopted measurement system, as discussed in a previous paper [57], has been evaluated using a reference parallel LCR circuit. This circuit consists of a 50 Ω resistive adapter, a fixed inductance, and a variable capacitance. The referenced values of the circuit impedance were measured with a Keysight E4980AL precision LCR meter. The estimated accuracy of the Vout/Vin ratio was better than ±0.2 dB in the interval from +5 to −25 dB and in the frequency range of 5 kHz to 1.5 MHz.

The SVM was implemented on a desktop computer (based on the Windows 10 × 64-bit operating system) using the open-source Python 3.7 from Anaconda [58]; the machine-learning algorithm was developed using the Scikit-learn library. Python is the programming language mostly used in artificial intelligence (AI) applications due to the availability of numerous libraries for continuous data acquisition and processing.

### 4.2. The Achieved Results

The proposed measurement technique is innovative and does not appear to have been tested by other authors. Due to the specificity of the acquired data (frequency response), there are no public datasets used by other authors against which to compare the performance of the proposed algorithm [59]. 

The measurement system was installed on a test facility, which was designed to generate electrical loads created by domestic users as part of the “non-intrusive infrastructure for monitoring loads in residential users” research project. The facility, located in the Electrical Engineering Laboratory of the University of L’Aquila (I), allows for the generation of electrical loads in a single or simultaneous way.

During the test phase, various parameters were evaluated in order to define the most significant sub-bands, the number of measurement points to be acquired, and the number of training examples needed to obtain a satisfactory performance. To this end, the precision, recall, and F1-Score during classification were evaluated [60]. These parameters were obtained using the numbers of true positive (TP), false positive (FP), true negative (TN), and false negative (FN) as follows:(8)$$Precision=\frac{TP}{TP+FP},$$
(9)$$Recall=\frac{TP}{TP+FN},$$
(10)$$F1-score=2\times \frac{precision\times recall}{precision+recall},$$

The concept of positive has been attributed to the ON state of household appliances and that of negative to the OFF state. Precision indicates all of the times the system has provided an indication of the ON state of an appliance and how many times the prediction has been correct. Precision does not take FNs into account. On the other hand, Recall indicates how many times the system has provided a correct indication about the ON state of the appliance compared to all of the samples in which the appliance was actually in the ON state. Recall does not take FPs into account. To have a metric capable of taking into account both FPs and FNs, the F1-Score is used, which is a harmonic mean of Precision and Recall.

Since, as already explained above, each appliance is associated with a SVM algorithm that reveals its presence, or not, the performance of each SVM was evaluated individually.

We started by acquiring 20 samples for each of the 24 scenarios, for a total of 480 training samples. Each sample consisted of an SFRA trace in which 200 points were acquired for each of the 3 sub-bands. Performance was evaluated on a test set consisting of 50 samples for each scenario, for a total of 1200 test samples. The obtained results, shown in Table 2, are already excellent, as 480 training samples is a relatively low number considering that acquiring a single sample takes about 40 s. The system does not make mistakes for five of the eight appliances analyzed and also shows high performance regarding the other three appliances. To define which of the three sub-bands made the most significant contribution to the identification of household appliances, the system’s performance was evaluated by providing the three sub-bands separately as input to the machine-learning system. The results are reported in Table 3 and a graphical comparison is provided in Figure 12.

In light of these results, it was decided that we would consider only the sub-bands of 10–100 kHz and 100 kHz–1 MHz in order to reduce the time required for the measurement. In fact, it is evident from Figure 12 that the 1–1.5 MHz band never allows for appliance discrimination that outperforms the previous bands. This reduces the time it takes to acquire a single trace to 22.56 s. Table 4 reports the performance evaluation using only the first two sub-bands as input.

Comparing the results with those of Table 2, it can be seen that the system’s performance has remained roughly unchanged. However, there is a significant improvement in the detection of the drill, highlighting that the 1–1.5 MHz sub-band introduced useless randomness for identification purposes. In this way, 400 points are acquired in the 10 kHz–1 MHz frequency band. 

The possibility of decreasing the number of acquired points has been evaluated. Therefore, in Table 5, the performances obtained for 200, 134, and 100 points are reported. Furthermore, Figure 13 shows a graphical comparison of the impact of the number of acquired points on the F1-Score.

The performance proved to be very good, even when only using 100 measurement points as a system input. In these conditions, in fact, the system made errors only for three of the eight appliances analyzed while maintaining a minimum F1-Score of 0.94. This reduction allowed a decrease in the execution time of the measurement system from 22.56 s to 6.09 s. The performances shown so far always foresaw 480 training samples (20 for each of the 24 scenarios). As a final analysis, the impact of the number of training samples on performance was evaluated as shown in Figure 14. Table 6 reports the results obtained using an SFRA trace consisting of 100 points acquired in the 10 kHz–1 MHz frequency band, reducing the number of samples used in the training phase.

The system maintains interesting performances even when trained with only one training sample for each scenario (therefore with 24 total training samples). This is mainly because the SVM natively suffers more from the quality of the training samples rather than the quantity, which is precisely because it builds a model based only on the most difficult samples to discriminate. 

Lower performance was found in the detection of the Lamp, Laptop, and Drill. In the case of the Lamp, this is due to the insignificance of its related load compared to the overall network, while in the case of the Laptop and Drill, it is due to the extreme variability of their working conditions. However, F1-Score values of 0.78, 0.87, and 0.94, respectively, can be considered largely satisfactory for a trained system with such a small number of samples.

In order to provide an overall assessment of the system’s performance, metrics widely used for multi-label classification systems were used, including micro-average and macro-average. As reported in (11)–(13), in the micro-average, all TPs, TNs, FPs, and FNs are summed for all of the labels and subsequently averaged:(11)$${Precision}_{micro-averaging}=\frac{\sum_{n=1}^{N}{TP}_{n}}{\sum_{n=1}^{N}{TP}_{n}+{FP}_{n}},$$
(12)$${Recall}_{micro-averaging}=\frac{\sum_{n=1}^{N}{TP}_{n}}{\sum_{n=1}^{N}{TP}_{n}+{FN}_{n}},$$
(13)$${F1-score}_{micro-averaging}=\frac{2\times {Precision}_{micro-averaging}\times {Recall}_{micro-averaging}}{{Precision}_{micro-averaging}+{Recall}_{micro-averaging}},$$

On the other hand, the macro-average, as reported in (14)–(16), is simply the average of the Precision and Recall for each label:(14)$${Precision}_{macro-averaging}=\frac{\sum_{n=1}^{N}{Precision}_{n}}{N},$$
(15)$${Recall}_{macro-averaging}=\frac{\sum_{n=1}^{N}{Recall}_{n}}{N},$$
(16)$${F1-score}_{macro-averaging}=\frac{2\times {Precision}_{macro-averaging}\times {Recall}_{macro-averaging}}{{Precision}_{macro-averaging}+{Recall}_{macro-averaging}},$$

The difference between the two lies is the fact that the micro-average reflects any imbalances in the dataset. Unbalance means there are test samples in a greater number of one or more classes than the others. In other words, having more samples for a given scenario, the macro-average, by creating a simple average of Precision, Recall, and F1-Score, does not consider this imbalance. On the contrary, the micro-average takes these situations into account.

In the case in question, the dataset is balanced; therefore, both averages are functional and adequate for verifying the performance of this system. Table 7 reports the micro-averages and macro-averages calculated based on the values reported in Table 6.

An additional consideration needs to be made to integrate the proposed system into an electrical system. As explained above, there is no interference with the normal operation of the devices during system operation. Furthermore, the system poses no problems to the EMI filters, which are the input stage of the monitored devices, as the powers involved—which can be associated with the test signal—are extremely low.

To analyze the operating conditions of the measurement system in detail, it was simulated in a SPICE environment.

Specifically, the simulation was oriented to analyze the effects produced by the test signal on commercial EMI filters that could be connected (to other devices) in proximity to the system being tested. The analysis was extended to the entire range of frequencies involved; as a reference, a commercial EMI filter family was considered [61] for standard use in commercial and residential apparatuses for AC currents up to 16 A_rms_ in single-phase systems. 

The analysis was extended to the entire range of frequencies involved. Figure 13 summarizes the scheme considered for the simulation. The resistance R_Load_ equal to 50 Ω was chosen in order to simulate the load of a generic household appliance (230 V_rms_/50 Ω = 4.6 A_rms_).

The system’s response was evaluated by varying the frequency in the range in which the proposed system operates in the final configuration (10 kHz–1 MHz). The frequency response of the current entering the EMI filter was evaluated. Several simulations were carried out by varying the RLC parameters of the EMI filter. The current was found to be harmless across the entire spectrum. As an example, Figure 15 shows the input current response obtained with the RLC parameters reported in Figure 16. The spectrum shows two resonance peaks and a maximum current draw of 4.64 mA.

The reduced value of this peak current does not lead to overheating of the filter components since the associated dissipated power is reduced. Furthermore, such verification is pejorative for the following reasons:

(1)The proposed system adopts a Digilent Analog Discovery 2 board, which has a limitation on the maximum output current that can be supplied by the DAC channels at 4 mA.(2)In our simulation, the measurement system is only connected to the device being tested. In the real case, the generator is connected to a generic socket of the electrical system; therefore, the current that can be supplied (4 mA) is distributed in the various parallel branches of the other connected devices, greatly reducing the intensity of the portion that could affect the EMI filters.

## 5. Conclusions and Final Remarks

Modern home, energy, and assisted environment management systems require nonintrusive monitoring of the power supply status of the various loads, regardless of information related to their consumption. This parameter is not easy to obtain from NILM systems. The SFRA technique, already widely used in the diagnostics of transformers and asynchronous motors, has been applied here to characterize household appliances from the point of view of their influence in modifying the frequency response of the electrical system. The obtained signature, influenced by the physical characteristics of the loads, has been used as input for a machine-learning algorithm, the SVM. The proposed algorithm has been implemented in Python’s open-source development environment, thus reducing the cost of the system.

A large campaign of measurements was carried out on a test facility, during which eight different electrical loads were powered individually and simultaneously. In particular, variable consumption loads, such as a drill and a laptop, were considered, which are generally among the most difficult for NILM systems to discriminate. The proposed system demonstrated excellent performance, even when trained with a minimum number of samples. In order to provide a comparison against other pre-published literature in the field, works that used similar metrics [62,63,64] were considered. The performances achieved by the cited works, by evaluating the F1-Score, were 91.5%, 93.2%, and 98.0%, respectively. The proposed system outperforms all three systems, as when all training data were provided (20 training samples for each scenario), the F1-Score achieved was 99.0%. It is important to note that the systems proposed in previous studies [62,63] were outperformed, even when the system was trained with the minimum number of samples when the system performance was 94.0%. The system is designed for local operation and is thus oriented toward edge implementation. The final system can be conveniently installed at any household outlet by detecting the presence of appliances connected to the system autonomously and providing data externally, for example, through wireless communication or the ability to download data histories via an SD card. The latter part will therefore be the subject of future research developments. Furthermore, the proposed system allows us to obtain information on which loads are powered in extremely short times (6.09 s in the final configuration of the system). These times were evaluated by considering both the time required to perform the measurement through the SFRA instrument and the time required to perform the prediction via the SVM classifier. Therefore, to ensure real-time operation, the edge system must incorporate multitasking capabilities. Two main tasks can be identified: in the first task, the system acquires and process the data to obtain the SFRA signature; in the second task, the system executes the SVM classifier and become ready to transfer the data over the WiFi network. The task of acquiring data and obtaining the signature, or SFRA trace, takes approximately 6 s, while the time required for processing the signature using the SVM classifier and transferring the data over WiFi (e.g., via an ESP32 module) is negligible and estimated to be around 10 ms based on experimental evaluations. This second task can be performed during the acquisition time of the first task. In fact, considering the first two signature-defining frequencies in the final configuration of the proposed system, namely 10,000 Hz and 11,350 Hz, the time needed for acquiring these initial points of the signature, as described in Section 2, amounts to 12 ms. Thus, under these conditions, the system can maintain real-time operation while meeting the requirements for post-processing and data transmission.

## Figures and Tables

**Figure 1 sensors-23-05226-f001:**
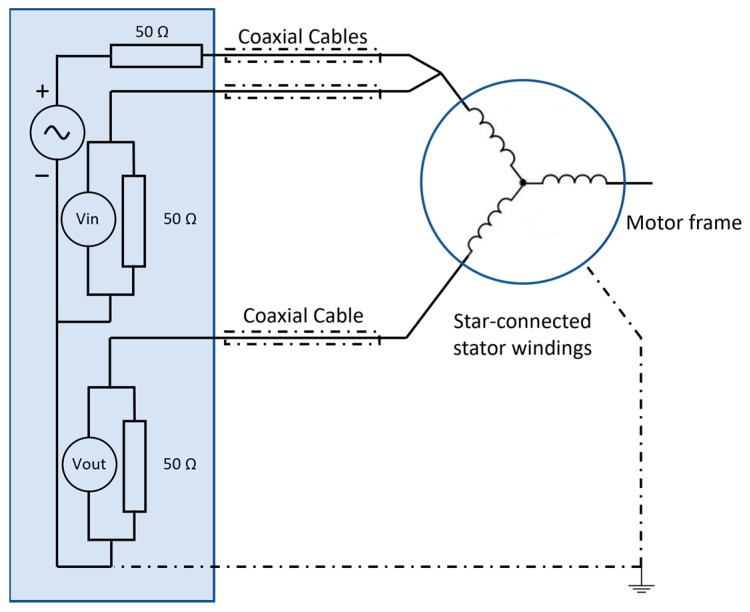
SFRA applied to a star-connected electric machine.

**Figure 2 sensors-23-05226-f002:**
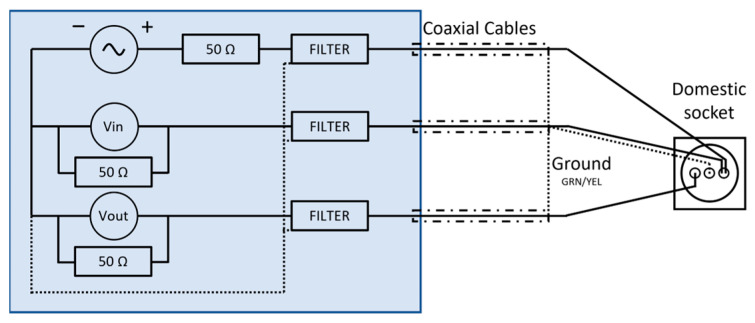
SFRA system.

**Figure 3 sensors-23-05226-f003:**
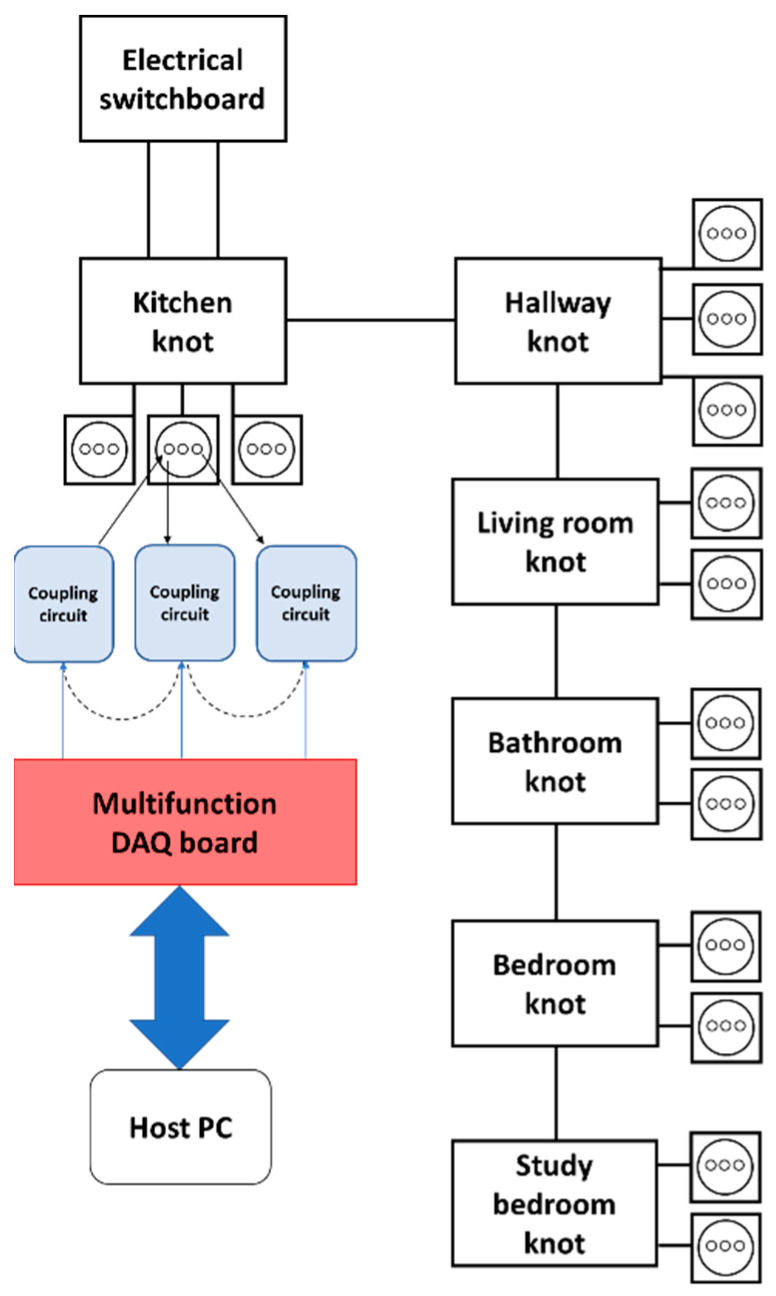
Installation of the SFRA in the test system.

**Figure 4 sensors-23-05226-f004:**
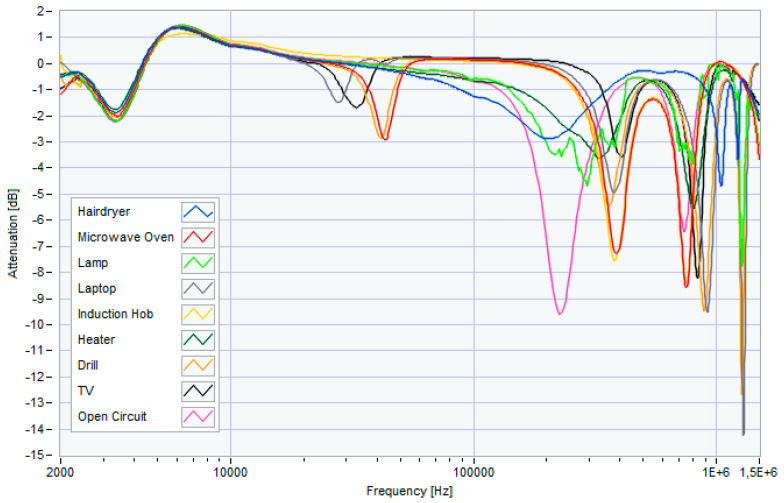
Frequency response of individually powered household appliances.

**Figure 5 sensors-23-05226-f005:**
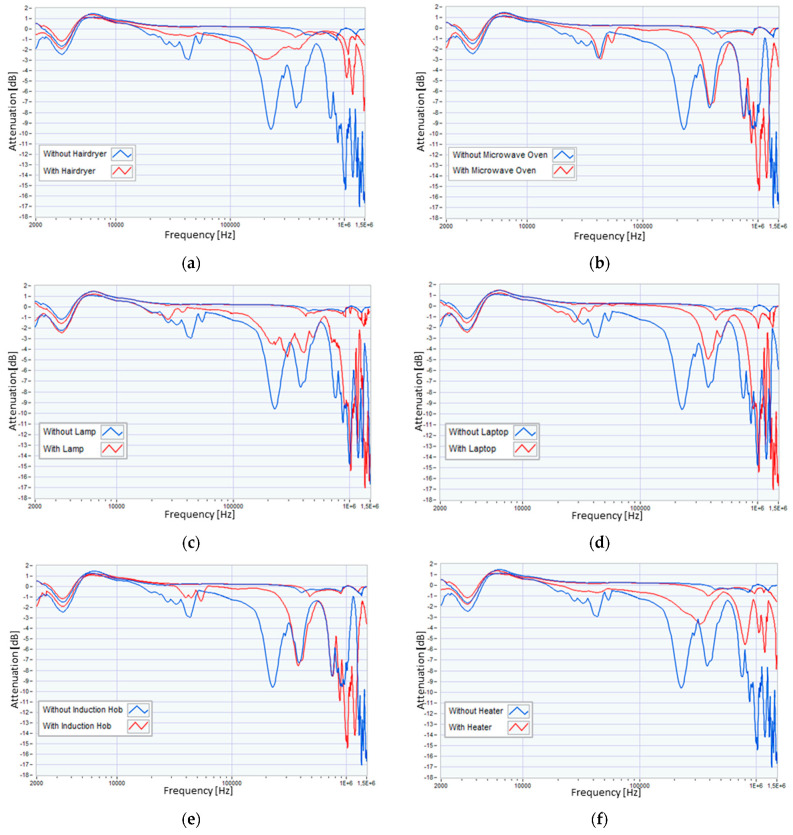
Envelopes of the traces obtained in the presence and absence of the: (**a**) hairdryer, (**b**) microwave oven, (**c**) lamp, (**d**) laptop, (**e**) induction hob, (**f**) heater, (**g**) drill, and (**h**) TV.

**Figure 6 sensors-23-05226-f006:**
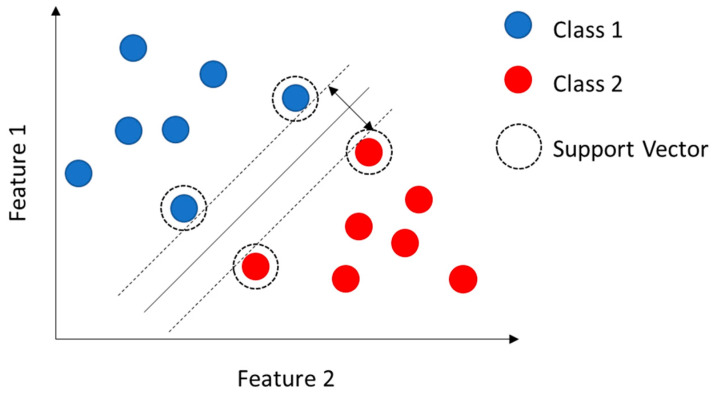
Representation of a linear classification problem in which the samples are defined by only two features.

**Figure 7 sensors-23-05226-f007:**
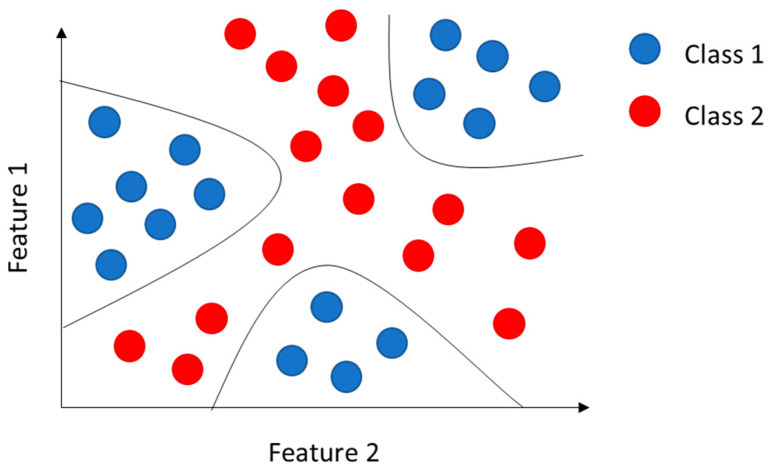
Representation of a non-linear classification problem in which the examples are defined by only two features.

**Figure 8 sensors-23-05226-f008:**
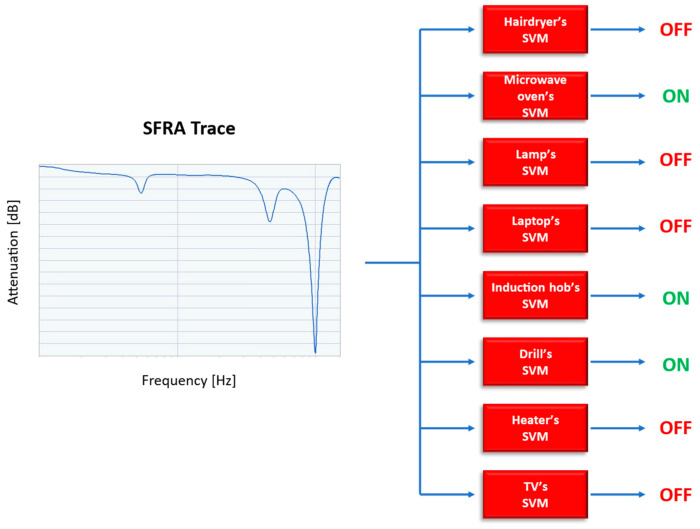
The proposed structure.

**Figure 9 sensors-23-05226-f009:**
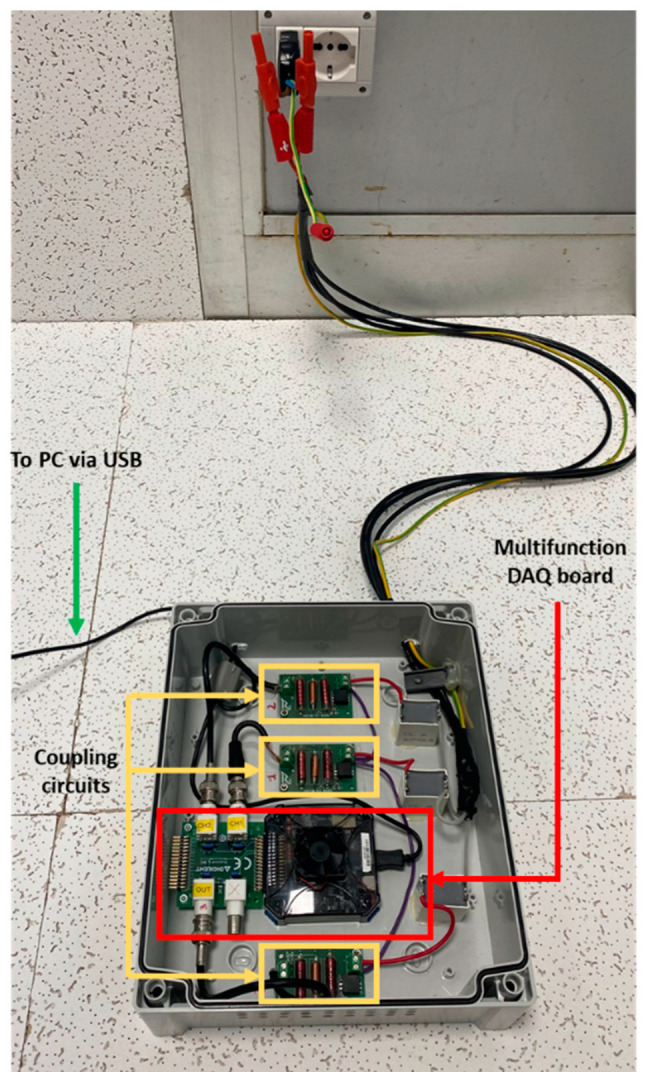
The SFRA measurement system.

**Figure 10 sensors-23-05226-f010:**
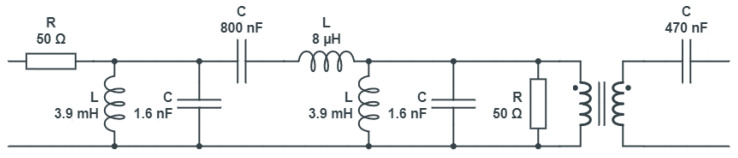
Coupling circuit for the signal generation section.

**Figure 11 sensors-23-05226-f011:**
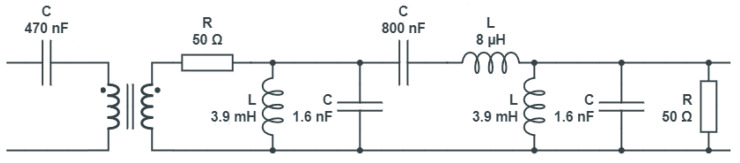
Coupling circuit for the signal acquisition section.

**Figure 12 sensors-23-05226-f012:**
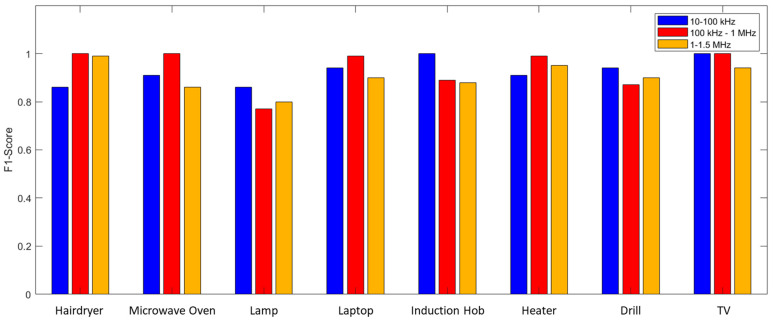
F1-Scores obtained for each considered sub-band.

**Figure 13 sensors-23-05226-f013:**
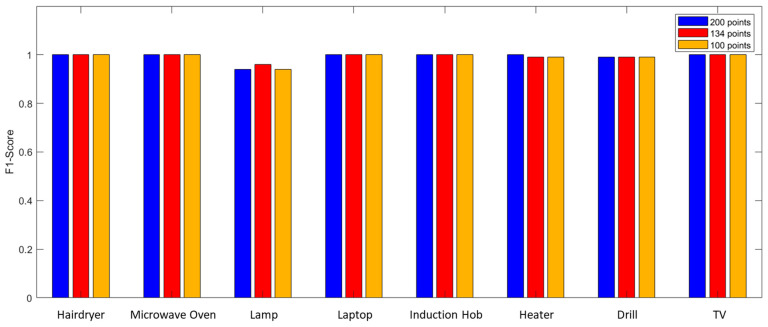
Graphical comparison of the impact of the number of acquired points on the F1-Score.

**Figure 14 sensors-23-05226-f014:**
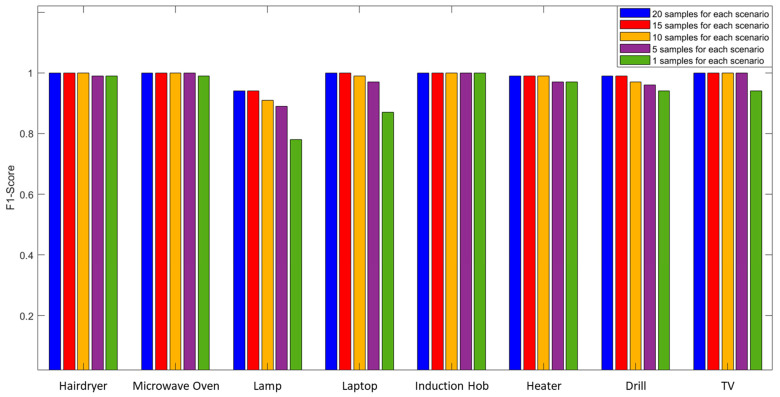
Graphical comparison of the impact of the number of training samples on the F1-Score.

**Figure 15 sensors-23-05226-f015:**
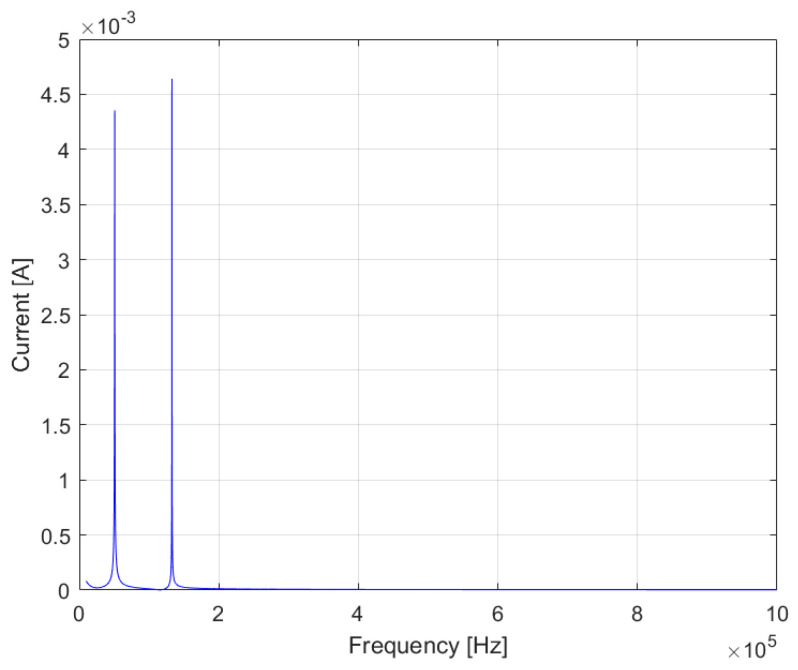
The frequency response of the input current to the EMI filter.

**Figure 16 sensors-23-05226-f016:**
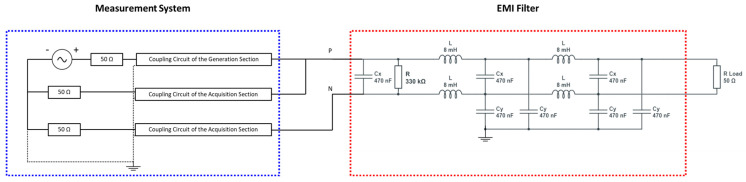
The scheme used for SPICE simulation.

**Table 1 sensors-23-05226-t001:** Power supply scenarios.

	Hairdryer	Microwave Oven	Lamp	Laptop	Induction hob	Heater	Drill	TV
**1**								
**2**	x							
**3**		x						
**4**			x					
**5**				x				
**6**					x			
**7**						x		
**8**							x	
**9**								x
**10**			x	x				
**11**	x					x		
**12**		x			x			
**13**			x	x				x
**14**	x					x		x
**15**		x			x			x
**16**			x	x			x	
**17**	x					x	x	
**18**		x			x		x	
**19**	x		x	x		x		
**20**		x	x	x	x			
**21**				x			x	x
**22**			x	x			x	x
**23**	x					x	x	x
**24**		x			x		x	x

**Table 2 sensors-23-05226-t002:** The results obtained with 480 training samples and 200 points for each sub-band.

	Total Errors	FP	FN	Precision	Recall	F1-Score
**Hairdryer**	0	0	0	1.00	1.00	1.00
**Microwave Oven**	0	0	0	1.00	1.00	1.00
**Lamp**	27	0	27	1.00	0.92	0.96
**Laptop**	0	0	0	1.00	1.00	1.00
**Induction Hob**	0	0	0	1.00	1.00	1.00
**Heater**	5	5	0	0.98	1.00	0.99
**Drill**	29	29	0	0.93	1.00	0.97
**TV**	0	0	0	1.00	1.00	1.00

**Table 3 sensors-23-05226-t003:** Performance evaluation for each sub-band.

10–100 kHz						
	Total Errors	FP	FN	Precision	Recall	F1-Score
**Hairdryer**	98	98	0	0.75	1.00	0.86
**Microwave Oven**	50	0	50	1.00	0.83	0.91
**Lamp**	110	96	14	0.78	0.96	0.86
**Laptop**	51	51	0	0.89	1.00	0.94
**Induction Hob**	0	0	0	1.00	1.00	1.00
**Heater**	61	61	0	0.83	1.00	0.91
**Drill**	48	48	0	0.89	1.00	0.94
**TV**	0	0	0	1.00	1.00	1.00
**100 kHz–1 MHz**						
	**Total Errors**	**FP**	**FN**	**Precision**	**Recall**	**F1-Score**
**Hairdryer**	0	0	0	1.00	1.00	1.00
**Microwave Oven**	0	0	0	1.00	1.00	1.00
**Lamp**	141	30	111	0.89	0.68	0.77
**Laptop**	9	0	9	1.00	0.98	0.99
**Induction Hob**	59	9	50	0.97	0.83	0.89
**Heater**	5	5	0	0.98	1.00	0.99
**Drill**	116	106	10	0.79	0.98	0.87
**TV**	0	0	0	1.00	1.00	1.00
**1–1.5 MHz**						
	**Total Errors**	**FP**	**FN**	**Precision**	**Recall**	**F1-Score**
**Hairdryer**	2	2	0	0.99	1.00	0.99
**Microwave Oven**	93	85	8	0.77	0.97	0.86
**Lamp**	115	0	115	1.00	0.67	0.80
**Laptop**	71	6	65	0.98	0.84	0.90
**Induction Hob**	79	74	5	0.80	0.98	0.88
**Heater**	29	29	0	0.91	1.00	0.95
**Drill**	90	76	14	0.84	0.97	0.90
**TV**	48	39	9	0.91	0.98	0.94

**Table 4 sensors-23-05226-t004:** The results obtained with 480 training samples and 200 points for each sub-band, using only the first two sub-bands.

	Total Errors	FP	FN	Precision	Recall	F1-Score
**Hairdryer**	0	0	0	1.00	1.00	1.00
**Microwave Oven**	0	0	0	1.00	1.00	1.00
**Lamp**	29	0	29	1.00	0.92	0.96
**Laptop**	4	4	0	0.99	1.00	0.99
**Induction Hob**	0	0	0	1.00	1.00	1.00
**Heater**	3	3	0	0.99	1.00	0.99
**Drill**	7	7	0	0.98	1.00	0.99
**TV**	0	0	0	1.00	1.00	1.00

**Table 5 sensors-23-05226-t005:** Performance evaluation as the points acquired decrease.

10 kHz–1 MHz (200 Points)						
	Total Errors	FP	FN	Precision	Recall	F1-Score
**Hairdryer**	0	0	0	1.00	1.00	1.00
**Microwave Oven**	0	0	0	1.00	1.00	1.00
**Lamp**	39	0	39	1.00	0.89	0.94
**Laptop**	0	0	0	1.00	1.00	1.00
**Induction Hob**	0	0	0	1.00	1.00	1.00
**Heater**	2	2	0	0.99	1.00	1.00
**Drill**	5	5	0	0.99	1.00	0.99
**TV**	0	0	0	1.00	1.00	1.00
**10 kHz–1 MHz (134 Points)**						
	**Total Errors**	**FP**	**FN**	**Precision**	**Recall**	**F1-Score**
**Hairdryer**	0	0	0	1.00	1.00	1.00
**Microwave Oven**	0	0	0	1.00	1.00	1.00
**Lamp**	26	0	26	1.00	0.93	0.96
**Laptop**	0	0	0	1.00	1.00	1.00
**Induction Hob**	0	0	0	1.00	1.00	1.00
**Heater**	5	5	0	0.98	1.00	0.99
**Drill**	5	5	0	0.99	1.00	0.99
**TV**	0	0	0	1.00	1.00	1.00
**10 kHz–1 MHz (100 Points)**						
	**Total Errors**	**FP**	**FN**	**Precision**	**Recall**	**F1-Score**
**Hairdryer**	0	0	0	1.00	1.00	1.00
**Microwave Oven**	0	0	0	1.00	1.00	1.00
**Lamp**	42	0	42	1.00	0.88	0.94
**Laptop**	0	0	0	1.00	1.00	1.00
**Induction Hob**	0	0	0	1.00	1.00	1.00
**Heater**	6	6	0	0.98	1.00	0.99
**Drill**	2	2	0	0.99	1.00	0.99
**TV**	0	0	0	1.00	1.00	1.00

**Table 6 sensors-23-05226-t006:** Performance evaluation as training samples decrease.

10 kHz–1 MHz (100 Points. 15 Samples for Each Scenario)						
	Total Errors	FP	FN	Precision	Recall	F1-Score
**Hairdryer**	0	0	0	1.00	1.00	1.00
**Microwave Oven**	0	0	0	1.00	1.00	1.00
**Lamp**	42	0	42	1.00	0.88	0.94
**Laptop**	0	0	0	1.00	1.00	1.00
**Induction Hob**	0	0	0	1.00	1.00	1.00
**Heater**	6	6	0	0.98	1.00	0.99
**Drill**	3	3	0	0.99	1.00	0.99
**TV**	0	0	0	1.00	1.00	1.00
**10 kHz–1 MHz (100 Points. 10 Samples for Each Scenario)**						
	**Total Errors**	**FP**	**FN**	**Precision**	**Recall**	**F1-Score**
**Hairdryer**	0	0	0	1.00	1.00	1.00
**Microwave Oven**	0	0	0	1.00	1.00	1.00
**Lamp**	59	0	59	1.00	0.83	0.91
**Laptop**	2	0	2	1.00	0.99	0.99
**Induction Hob**	0	0	0	1.00	1.00	1.00
**Heater**	5	5	0	0.98	1.00	0.99
**Drill**	23	21	2	0.95	1.00	0.97
**TV**	0	0	0	1.00	1.00	1.00
**10 kHz–1 MHz (100 Points. 5 Samples for Each Scenario)**						
	**Total Errors**	**FP**	**FN**	**Precision**	**Recall**	**F1-Score**
**Hairdryer**	5	5	0	0.98	1.00	0.99
**Microwave Oven**	0	0	0	1.00	1.00	1.00
**Lamp**	71	0	71	1.00	0.80	0.89
**Laptop**	26	0	26	1.00	0.94	0.97
**Induction Hob**	0	0	0	1.00	1.00	1.00
**Heater**	16	16	0	0.95	1.00	0.97
**Drill**	31	29	2	0.93	0.99	0.96
**TV**	0	0	0	1.00	1.00	1.00
**10 kHz–1 MHz (100 Points. 1 Sample for Each Scenario)**						
	**Total Errors**	**FP**	**FN**	**Precision**	**Recall**	**F1-Score**
**Hairdryer**	5	5	0	0.98	1.00	0.99
**Microwave Oven**	1	1	0	0.99	1.00	0.99
**Lamp**	125	0	125	1.00	0.64	0.78
**Laptop**	95	2	93	0.99	0.77	0.87
**Induction Hob**	0	0	0	1.00	1.00	1.00
**Heater**	17	17	0	0.95	1.00	0.97
**Drill**	53	53	0	0.88	1.00	0.94
**TV**	8	8	0	0.98	1.00	0.94

**Table 7 sensors-23-05226-t007:** Impact of the size of the training set on multi-label classification.

	Micro-Average			Macro-Average		
Training Samples for Each Scenario	Precision	Recall	F1-Score	Precision	Recall	F1-Score
**20**	0.99	0.98	0.99	0.99	0.98	0.99
**15**	0.99	0.98	0.99	0.99	0.98	0.99
**10**	0.99	0.97	0.98	0.99	0.97	0.98
**5**	0.98	0.96	0.97	0.98	0.96	0.97
**1**	0.96	0.92	0.94	0.97	0.92	0.94

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
