# Peer review of "A New NILM System Based on the SFRA Technique and Machine Learning"

_sensors, 2023, doi:10.3390/s23115226_

Round 1
Reviewer 1 Report
This paper presents a new NILM system based on SFRA and ML. This new system has been implemented on the Analog Discover 2 NI Edition Evaluation Board to generate the SFRA and capture the resulting signal for further classification using an SVM. The results show that a frequency sweep would allow a wider range of devices to be identified with greater accuracy, and it can be seen from the frequency sweep that differences between devices appear from 10 kHz.
The article is well written and easy to follow, although there are aspects that I think should be improved:
- In the abstract there should be some comment on the results obtained in the article, not just that they are illustrated and discussed.
- In the introduction there is a review of the state of the art, but I think that this review is very limited and almost 40% are references from the authors themselves. I believe that a more thorough review of the state of the art should be carried out, including a new section with a review of previous work.
- The article is well written, but the place where the figures and tables appear makes me lose a little the reference of what I am reading, because the figure has appeared before or appears later. I suggest that the location of the figures and tables be considered, as well as the following changes to the figures.
* Figures 4 and 5. Lettering of axis labels and larger legend.
* Table 1. Smaller.
* Figure 10. Larger or better distributed so that the labels can be read.
- Lines 146, 155 and 225 stand alone and are information attached to the surrounding paragraphs, please consider adding them with the rest of the information.
- Section 2 introduces the SFRA and the one to be used for this work, but does not provide much detail on the subsystem until the SPICE analysis is performed in the results. I believe that these design details can be included in Section 2 and more details on filters etc. can be provided. In summary, I think this section should better explain the SFRA.
- Lines 172-173 Correct the OFDM acronym as they appear unrelated.
- From line 177 onwards we start talking about the measurements made and show frequency sweep figures. I think this is related to the SFRA system, but the results of the experiments are shown. The information can be separated in this section and more details about the experiments and the relationship to Table 1 should be given in this section.
- Section 3 provides an overview of SVM that can be included in the introduction or review of previous work. In addition, more information about the SVM architecture used for this work should be provided.
- Section 4 on experimental results shows the equipment developed, but I believe that a more detailed outline of the SVM architecture should be provided in this section.
- A schematic of the LabVIEW programme used should be included in this section.
- Although the authors indicate this at the beginning of section 4.1. The data used to train and validate the SVM is the same, making it difficult to replicate or validate for the rest of the scientific community. Have you considered publishing the database you have created for the scientific community?
- Regarding the results shown, they are poorly commented, I think a deeper analysis of the results should be done. In addition, line 427 says "in light of these results", but the reason for this decision has not been justified.
- On the other hand, in Table 6 and Table 7, there is only an analysis of sample reduction in the face of training, but this is not clear in the text. Please clarify this part and how it relates to or affects the proposed system.
- I think that a comparison with other works should be made, even if they do not have the same device, but there are previous works that try to analyse with a NILM system the same devices as yours, is it possible to add a comparative table and a discussion of the results?
- Finally, I think you should be more detailed and in-depth in the conclusion.
Reviewer 2 Report
This article has been written well but the author needs to explain the before queries for the next level of proceeding.
1. The numerical summarization of the results is to be included in the abstract.
2. The novelty of the article is to be strengthened and found to be weak.
3. In an article related to Machine Learning the author has only provided a basic introduction about the machine learning part. Here in this article basic or the introduction of the machine learning part is not in need.
4. The process of the SVM defined in this article is not clear and is clumsy.
5. The article is poor in mathematical modeling, the author can look into this and check for the mathematical modeling, this mathematical modeling related to ML/DL can be provided in any domain but it needs to fulfill. https://doi.org/10.3390/electronics11244178
6. Comparative or the graphical output provided here are not up to the mark, you can refer to the above article link and provide a few more comparative outputs in your article.
Reviewer 3 Report
The paper addresses a topic of interest, I have a few comments to make though.
1. Please improve the references list by both a) reducing the amount of your papers that you are including there (self-citation is extremely high) and b) adding more NILM-related references that are recently published. An example: Athanasiadis, Christos L., Theofilos A. Papadopoulos, and Dimitrios I. Doukas. "Real-time non-intrusive load monitoring: A light-weight and scalable approach." Energy and Buildings 253 (2021): 111523.
2. Benchmarking against other pre-published literature in the field is needed. That will help for the contributions to become more clear.
3. Please improve the use of the english language, there are many points that the document can improve significantly.
4. A discussion section regarding the computational burden/effort of this implementation is needed. Working with such high grannularities of data will make the final solution really heavy and intensive.
5. What about the dataset that you have used to generate these results? Could the authors make that publicly available? Do the authors consider the dataset big enough for important conclusions to be drawn?
6. Where is such a solution expected to run? On the edge or maybe on the cloud?
--
Round 2
Reviewer 1 Report
Thank you for considering most of the comments in the previous review.
I think a strength of this article is the use of high sampling frequencies to identify loads, as it allows differences other less conventional and classical ones. However, this requires a high computational burden, even more so if SVM based algorithms are added. With all this, what bandwidth should the system require to be used in real time? The conclusions indicate that it would be sent by wireless communication or storage in SD, but this depends on whether only the event is sent or periodically the acquired signal.
It is my opinion that from line 636 to 645 certain aspects of the final system should be improved, since a series of future works can be indicated, instead of providing general aspects of conventional systems, but forgetting key aspects that make edge computing necessary, but at the same time so difficult to carry out.
Reviewer 2 Report
The author has done well in this revision but needs to clarify the comments below in the response report.
1. The overall percentage mentioned in the outcomes seems to be 9 percent. The author needs to clarify or send the code how these higher percentage has been received.
2. In the Precision and Recall most of the values have touched 1, Author needs to clarify the values supplied for getting a higher percentage.
3. Since the percentage is high, the author can send the ROC in the Reviewer report.
Reviewer 3 Report
no further comments
Author Response
The authors would like to thank this reviewer.
Round 3
Reviewer 2 Report
The comments sent by the author has accepted. Getting more than 99 percent accuracy and RoC near to 1 is more interesting.
Can the author upload the code in the GitHub repository or in 'Data in Brief', to test for the reachability of getting more than 99 percent?
Author Response
Certainly, the code will be made available on a Github repository along with the dataset.